# Effects of Outdoor and Household Air Pollution on Hand Grip Strength in a Longitudinal Study of Rural Beijing Adults

**DOI:** 10.3390/ijerph22081283

**Published:** 2025-08-16

**Authors:** Wenlu Yuan, Xiaoying Li, Collin Brehmer, Talia Sternbach, Xiang Zhang, Ellison Carter, Yuanxun Zhang, Guofeng Shen, Shu Tao, Jill Baumgartner, Sam Harper

**Affiliations:** 1Department of Epidemiology, Biostatistics and Occupational Health, McGill University, Montreal, QC H3A 1G1, Canada; wenlu.yuan@mail.mcgill.ca (W.Y.); talia.sternbach@mail.mcgill.ca (T.S.); 2Department of Mechanical Engineering, Colorado State University, Fort Collins, CO 80523, USA; xiaoying.li@colostate.edu; 3Department of Civil and Environmental Engineering, Colorado State University, Fort Collins, CO 80523, USA; collin.brehmer@colostate.edu (C.B.); ellison.carter@colostate.edu (E.C.); 4Department of Geography, McGill University, Montreal, QC H3A 0B9, Canada; xiang.zhang7@mail.mcgill.ca; 5Beijing Yanshan Earth Critical Zone National Research Station, College of Resources and Environment, University of Chinese Academy of Sciences, Beijing 100049, China; yxzhang@ucas.edu.cn; 6Laboratory for Earth Surface Processes, SinoFrench Institute for Earth System Science, College of Urban and Environmental Sciences, Peking University, Beijing 100871, China; gfshen12@pku.edu.cn (G.S.); taos@pku.edu.cn (S.T.); 7Department of Equity, Ethics, and Policy, McGill University, Montreal, QC H3A 1G1, Canada

**Keywords:** household air pollution, outdoor air pollution, grip strength, rural population

## Abstract

**Background:** Outdoor and household PM_2.5_ are established risk factors for chronic disease and early mortality. In China, high levels of outdoor PM_2.5_ and solid fuel use for cooking and heating, especially in winter, pose large health risks to the country’s aging population. Hand grip strength is a validated biomarker of functional aging and strong predictor of disability and mortality in older adults. We investigated the effects of wintertime household and outdoor PM_2.5_ on maximum grip strength in a rural cohort in Beijing. **Methods:** We analyzed data from 877 adults (mean age: 62 y) residing in 50 rural villages over three winter seasons (2018–2019, 2019–2020, and 2021–2022). Outdoor PM_2.5_ was continuously measured in all villages, and household (indoor) PM_2.5_ was monitored for at least two months in a randomly selected ~30% subsample of homes. Missing data were handled using multiple imputation. We applied multivariable mixed effects regression models to estimate within- and between-individual effects of PM_2.5_ on grip strength, adjusting for demographic, behavioral, and health-related covariates. **Results:** Wintertime household and outdoor PM_2.5_ concentrations ranged from 3 to 431 μg/m^3^ (mean = 80 μg/m^3^) and 8 to 100 μg/m^3^ (mean = 49 μg/m^3^), respectively. The effect of a 10 μg/m^3^ within-individual increase in household and outdoor PM_2.5_ on maximum grip strength was 0.06 kg (95%CI: −0.01, 0.12 kg) and 1.51 kg (95%CI: 1.35, 1.68 kg), respectively. The household PM_2.5_ effect attenuated after adjusting for outdoor PM_2.5_, while outdoor PM_2.5_ effects remained robust across sensitivity analyses. We found little evidence of between-individual effects. **Conclusions**: We did not find strong evidence of an adverse effect of household PM_2.5_ on grip strength. The unexpected positive effects of outdoor PM_2.5_ on grip strength may reflect transient physiological changes following short-term exposure. However, these findings should not be interpreted as evidence of protective effects of air pollution on aging. Rather, they highlight the complexity of air pollution’s health impacts and the value of longitudinal data in capturing time-sensitive effects. Further research is needed to better understand these patterns and their implications in high-exposure settings.

## 1. Background

Outdoor and household (indoor) fine particulate matter (PM_2.5_) air pollution are well-established risk factors for a wide range of adverse health outcomes, including cardiorespiratory diseases, neurotoxicity, and declines in physical and cognitive function [1]. In rural China, exposure to household PM_2.5_ remains widespread, with an estimated 40% and 85% of households using solid fuel (i.e., coal and biomass) for cooking and heating, respectively [2]. During winter, increased solid fuel combustion for space heating and unfavorable meteorological inversions contribute to prolonged periods of poor air quality and higher PM_2.5_ exposures [3,4] and health risks [5].

As China’s population ages, there is increasing public health interest in identifying early indicators of functional decline influenced by environmental exposures such as air pollution. These indicators can help detect individuals at risk of poor health outcomes, allowing for timely interventions to prevent disability and preserve quality of life. Hand grip strength is a validated, low-cost, and non-invasive measure of physical functioning that predicts future disability, hospitalization, and mortality among older adults [6,7,8,9,10]. Declines in grip strength may reflect subclinical changes in the musculoskeletal or neurological systems that are susceptible to environmental stressors such as air pollution. Though the biological mechanisms linking air pollution to grip strength are not fully understood, plausible pathways include air pollution-induced oxidative stress and systemic inflammation, which can impact muscle mass and strength [11,12,13,14]; insulin resistance, obesity, and neurodegeneration (e.g., Parkinson’s and Alzheimer’s disease), which are linked to impaired muscle function [15,16,17]; and frailty and physical disability, which can lead to loss of muscular strength [18,19]. Thus grip strength could serve as a useful indicator for assessing the broader health impacts of air pollution, especially in rural settings of solid fuel use where both population aging and exposures to air pollution are pronounced.

A longitudinal study in China observed greater reductions in grip strength among individuals using solid fuel for cooking compared to those who used clean fuel (2.27 versus 1.67 kg reduction over four years [20]). Other longitudinal studies in China reported increased risk of sarcopenia, i.e., age-related muscle mass loss assessed through physical performance and grip strength, among participants in homes using solid fuel compared with participants using clean fuel (hazard rate ranging from 1.26 to 1.56 [21,22,23]). Notably, these longitudinal studies excluded nearly half of the baseline participants due to missing data on household fuel use or grip strength, which may have introduced selection bias if exclusions were conditioned on variables related to both of these variables. Several cross-sectional studies conducted in countries including China also reported inverse associations between household and outdoor air pollution and small decreases in grip strength (0.07 or 0.69 kg reduction per 3.9 or 10 μg/m^3^ in annual outdoor PM_2.5_, respectively [24,25]). However, these studies relied on surrogate exposure measures (e.g., household fuel type or satellite-derived PM_2.5_) rather than direct measurements in homes or villages, and were primarily conducted in cities where air pollution levels, sources, and chemical composition differ substantially from those in rural areas [26,27]. Further, the cross-sectional design limits causal inference and does not capture temporal variation in air pollution, which may be especially relevant in settings like China with large annual and seasonal fluctuations in PM_2.5_ [28,29].

To address these limitations, we analyzed longitudinal, individual-level measurements of outdoor and household (indoor) PM_2.5_ and repeated assessments of grip strength in a rural Beijing cohort. Our aim was to assess the impact of wintertime outdoor and household PM_2.5_ on grip strength among middle-aged and older adults living in areas where solid fuel combustion is a major source of air pollution. Improving our understanding of how air pollution affects physical function in high-exposure rural settings can help guide interventions to promote healthy aging and advance environmental health equity in rural China and similar settings globally.

## 2. Methods

### 2.1. Study Design and Data Collection

This study was conducted within the Beijing Household Energy Transition (BHET) study, which evaluates the air quality and health impacts of clean energy transition. A total of 1438 adults from 1236 households in 50 villages located in four rural Beijing districts (Fangshan, Huairou, Mentougou, and Miyun) were enrolled and followed between December 2018 to January 2022. Wintertime data collection occurred during three waves: 2018–2019 (wave 1), 2019–2020 (wave 2), and 2021–2022 (wave 4). Measurements in 2020–2021 (wave 3) were limited to environmental measurements due to COVID-19 distancing requirements and therefore excluded from this study.

The study protocols were approved by research ethics boards at Peking University (IRB00001052-18090), Peking Union Medical College Hospital (HS-3184), and McGill University (A08-E53-18B).

### 2.2. Study Population

Villages were selected for the study because they were eligible for, but not currently participating in, a rural clean heating policy in northern China, meaning that they were permanent villages and predominately used household coal stoves for winter heating (location of study villages is shown in Appendix A). Participants were eligible for the study if they were over 40 years old, primarily resided in the study village, had no plans to move within the next year, and were not currently undergoing immunotherapy or corticosteroid treatment. Although we initially intended to use a systematic stratified sampling method based on household rosters, this approach proved infeasible due to outdated records—many listed households were no longer occupied due to migration, death, or relocation to urban areas. Given time constraints (i.e., enrolling one village per day before the end of the heating season), we adopted a pragmatic sampling strategy in formal sampling: field staff were first introduced to several households (typically 4–5) by a village guide, and then recruited additional participants via referrals from enrolled households or by approaching residents encountered in the village. When necessary, the village guide provided further assistance with introductions. We recruited approximately 20 households in each village and, in each household, obtained a household roster. Our tablet-based survey incorporated a randomization tool that randomly ordered household occupants listed on the roster. We recruited a participant in each household by starting at the top of the randomly ordered list until an eligible participant was identified. In follow-up visits to the study villages, staff first approached households with participants from an earlier wave. If a previous participant was not at home or refused to participate, staff first tried to randomly recruit the next eligible participant listed on the randomized household roster. If there was not another eligible or willing participant in the same household, we recruited a participant from a new household using the same process for household and participant selection described above. Details about participant recruitment can be found elsewhere [30].

Of the 1438 enrolled participants, this study includes 877 participants that had at least two grip strength measurements across the three waves, yielding 2176 observations (a flowchart of selection into this study is shown in Appendix A).

### 2.3. Air Pollution Measurements

#### 2.3.1. Measurement of Household (Indoor) PM_2.5_

In wave 2, we randomly selected 6 households from the ~20 recruited households in each village for the measurement of household PM_2.5_. In wave 4, we aimed to monitor household PM_2.5_ in the same households sampled in wave 2. If household occupants were not at home or if participants declined household PM_2.5_ monitoring, we randomly recruited another household already enrolled in the study (n = 52). In total, household (indoor) PM_2.5_ was continuously measured for at least two months in the randomly selected subsample of 300 participant households in waves 2 and 4 using real-time PM_2.5_ sensors (PMS7003 Plantower, Zefan, Inc., Tianjin, China) with a counting efficiency of 98% for particles > 0.5 μm in diameter [31]. Sensors were placed on an elevated surface in the room where participants reported spending most of their time while awake. Household PM_2.5_ measurements typically started between December and January and lasted for three to four months. We co-located the sensors with gravimetric filter-based instruments in 50% of homes in each village for the first 24 h of measurement. The sensor-based measurements were then calibrated using the slopes from established linear regressions between the filter-based PM_2.5_ mass concentrations (i.e., the reference concentrations) and the sensor-based PM_2.5_ concentrations averaged over the same sampling period as the filter-based samples. Details on gravimetric calibration can be found elsewhere [32].

For each sampled household, we calculated the average wintertime household PM_2.5_ concentration collected from 15 January and 15 March, which falls within the winter heating period, to maximize the consistency in the measurement period across participant homes. Due to power shortages or sensor damage, some sensors (12% in wave 2 and 19% in wave 4) did not monitor household PM_2.5_ during the entire two-month period and were excluded from the analysis.

#### 2.3.2. Measurement of Outdoor (Community) PM_2.5_

Outdoor PM_2.5_ was measured continuously in the study villages from December 2018 to March 2019 in wave 1, from November 2019 to August 2020 in wave 2, and from November 2021 to July 2022 in wave 4, which covers the entire winter heating period; details are described elsewhere [33]. Briefly, we placed one set of air monitors near the village center and a second set at least 500 m away. Monitors were placed at least 1.5 m above the ground and away from visible direct sources of air pollution (e.g., chimneys, small-scale industry). We conducted real-time measurements using the same sensors as those used for the indoor measurements. An ultrasonic personal aerosol sampler for filter-based PM_2.5_ measurement was co-located with the sensors in rotation in each village. We applied a sensor-specific adjustment factor using linear regressions between the filter-based and the sensor-based PM_2.5_ concentrations averaged over the same measurement period.

### 2.4. Covariates

In each wave, trained staff conducted face-to-face interviews (median time = 60 min) with participants during household visits, and administered structured questionnaires that collected information on potential confounders and predictors of grip strength including age, sex, education, occupation, marital status, tobacco exposure, smoking intensity, alcohol consumption, farming frequency, exercise frequency, medical history (e.g., hypertension, diabetes, stroke), household income, and household-owned assets. Interviews were conducted in Mandarin Chinese, and their responses were recorded on tablets. Questionnaire data were reviewed daily by field supervisors to evaluate completeness and quality. Detailed information about the survey testing and quality control measures are provided elsewhere [30].

Waist circumference, which is considered an indicator of cardiovascular disease risk and obesity [34], was measured with a tape measure. Participants were asked to stand and breathe normally while staff placed the tape measure midway between the top of their hip bones and the bottom of their ribs, in line with their belly button, and wrapped it around their waist loose enough to fit one finger inside the tape. Indoor temperature was measured using a digital thermometer (Tianjianhuayi Inc., Beijing, China) during household visits.

An asset-based wealth index was calculated based on household owned assets and income using principal component analysis. Details on the wealth index calculation can be found elsewhere [33].

### 2.5. Grip Strength Measurements

Grip strength was measured using a JAMAR hydraulic hand dynamometer (Performance Health, Downer Grove, IL, USA), which has been described as the gold-standard tool for handgrip strength evaluations [35,36], following the recommendations for standard clinical assessment [37]. Briefly, participants were seated in an upright position with their arms along their body; the arm was bent at 90° at the elbow with the forearm and wrist in the neutral position. Participants were instructed to squeeze the grasp of the dynamometer as hard as they could. As is typical of grip strength measurement protocols, study staff encouraged all participants during measurement and held the readout dial to prevent inadvertent dropping during the test. Grip strength (in kg of force) was measured three times in each hand with brief pauses between each measurement. The highest value from each wave was used in the statistical analysis.

### 2.6. Multiple Imputation for Missing Data

We applied multiple imputation by chained equations (MICE) [38] to impute missing household PM_2.5_ in wave 1, as well as for households not sampled for measurements in waves 2 and 4. MICE was also used to impute missing covariates. Missingness accounted for 4% of all observations.

Imputation models included household-level characteristics (e.g., heating area, insulation status), village-level characteristics (wintertime outdoor temperature and dew point from 15 January to 15 March), participant socio-demographic variables (e.g., highest education obtained, current occupation), and air pollution (seasonal outdoor and household PM_2.5_ and 24 h personal exposure to PM_2.5_; details about 24 h personal exposure are described in Appendix A: The results for 24 h personal exposure to PM_2.5_ concentration measurements). These variables were selected based on their potential associations with household PM_2.5_.

A total of 30 imputations were performed, each with 30 iterations, using a 50 × 50 predictor matrix derived from the selected variables (selected variables, the number of missing, and their proportions in the datasets are shown in Appendix A). Convergence was assessed via trace plots (Appendix A). Descriptive statistics comparing observed only versus observed plus imputed household PM_2.5_ are presented in Appendix A.

Multiple imputation was implemented using the *mice* package (version 3.16.0) in R [39].

### 2.7. Statistical Analysis

We estimated the effects of outdoor and household PM_2.5_ on grip strength using multivariable mixed effects regression models with participant-specific random intercepts to account for repeated measurements. To distinguish between- and within-individual effects, we included both participant-mean PM_2.5_ and mean-centered PM_2.5_ values as predictors. The models were adjusted for sex, age, marital status, highest education obtained, current occupation, exposure to tobacco smoke, smoking intensity (typical number of cigarettes consumed per day if reported to be current smokers), frequency of alcohol consumption, frequency of farming, exercise frequency, self-reported health status, waist circumference (modeled as a natural cubic spline with 2 degrees of freedom, *df*), and asset-based wealth index quartile. The models were run across 30 imputed data frames and the results were pooled using Rubin’s rules [38]. We used the following model structure:$$GS_{it}=\gamma_{00}+\beta_{1b}{\bar{AP}}_{it}+\beta_{1w}({{AP}_{it}-\bar{AP}}_{it})+\beta_{2}C_{it}+\beta_{3}C_{i}+\mu_{0i}+\varepsilon_{it}\;\mu_{0i}\sim N(0,\sigma_{i}^{2})$$
where $GS_{it}$ is the maximum grip strength for individual *i* in study wave *t*; $\beta_{1w}$ represents the effect of a 10 μg/m^3^ mean-centered within-individual change in PM_2.5_ on maximum grip strength; $\beta_{1b}$ represents the effect of a 10 μg/m^3^ time-averaged between-individual change in PM_2.5_ on maximum grip strength; $C_{it}$ and $C_{i}$ represent the set of individual time-varying (e.g., exercising frequency) and time-invariant covariates (e.g., sex); $\mu_{0i}$ represents the individual-specific random effect which follows a normal distribution $\sim N(0,\;\sigma_{i}^{2})$; $\varepsilon_{it}$ represents the residual idiosyncratic error; and $\gamma_{00}$ represents the model intercept, indicating the overall average maximum grip strength in the study population.

We included an individual-level mean-centered PM_2.5_ term (${{AP}_{it}\;-\;\bar{AP}}_{it}$) to estimate within-individual effects of PM_2.5_ on maximum grip strength (i.e., the “within-individual” effect [40]; $\beta_{1w}$). Conditioning on an individual’s mean exposure across three waves (${\bar{AP}}_{it}$) controls for observed and unobserved time-invariant individual level confounders (e.g., genetics). These estimates assume that time-varying confounders are measured accurately, with no systematic bias from measurement error or selection bias [41]. A time-averaged, fixed effect term (${\bar{AP}}_{it}$) was used to estimate the effects of a 10 μg/m^3^ between-individual change in PM_2.5_ on participant’s maximum grip strength (i.e., the “between-individual” effect [40]; $\beta_{1b}$). By estimating both the within-individual and between-individual relationships between air pollution and grip strength, we distinguish two sources of variation in PM_2.5_. The within-individual effect reflects how changes of PM_2.5_ in the same individual over time are associated with changes in grip strength and are not biased by time-invariant factors. In contrast, between-individual effect captures how differences in average PM_2.5_ exposure across individuals relate to grip strength, and may still be confounded by unmeasured time-fixed factors.

We evaluated potential nonlinearity between grip strength and continuous independent variables using natural cubic splines with 2–4 df and excluding imputed data, resulting in 389 observations from 265 participants. Evidence of nonlinearity was observed only for waist circumference (Appendix A); therefore, we modeled it using a spline with 2 df in all models after observing no difference in models with 2, 3, or 4 df. The effects of PM_2.5_ on grip strength were mostly linear, except for a potential flattening at very high household PM_2.5_ levels. However, this pattern disappeared after excluding the top 3% of observations (>257.7 μg/m^3^), and the estimates stabilized, supporting a linear specification in our main models (Appendix A). We produced diagnostic plots to confirm that the model assumptions held (Appendix A).

The fully adjusted household PM_2.5_ model also adjusted for outdoor PM_2.5_ to better isolate the effects of indoor sources. We also examined interactions between PM_2.5_ and sex, age, highest education, current occupation, exposure to tobacco smoke, drinking frequency, farming frequency, exercising frequency, or wealth index quartile by including product terms between each potential effect modifier and exposure. We tested the models with and without interaction terms using F-test and plotted the marginal effects of air pollution by different subgroups to highlight observed interactions.

We conducted numerous sensitivity analyses to evaluate the robustness of our findings. First, we restricted the analysis to complete cases (no imputed data), limited to waves 2 and 4 when household PM_2.5_ was measured. Second, we added a random intercept for village of residence to account for potential village-level clustering. Third, we additionally adjusted for co-morbid conditions (e.g., hypertension, COPD, TB, diabetes, asthma, hepatitis, cirrhosis, kidney disease, rheumatoid arthritis, heart disease, stroke) and recent hospitalization by adding a categorical variable to indicate the number of conditions. These conditions could be confounders [9,42,43,44,45,46] but also along the PM_2.5_–grip strength pathway [1]. Fourth, we additionally adjusted for indoor temperature measured during household visits, which may affect both heating behavior and muscle strength but was excluded from the main model due to limited evidence of these effects [47,48]. Fifth, we additionally adjusted for the clean energy policy implementation status of participant’s living village for the household PM_2.5_ model, considering its impact on wintertime household PM_2.5_ concentration [30] and potential influence on muscle strength through other factors other than air pollution. Sixth, we excluded observations with household PM_2.5_ measurements in the highest 3% and 5% to assess the influence of extreme values. Seventh, we excluded observations with household PM_2.5_ measurements less than 20% (<2 weeks) of the 2-month study period (n = 25) to minimize measurement error of exposure. Eighth, we limited the analysis to waves 1 and 2, which were both conducted before the start of the COVID-19 pandemic, to rule out confounding related to pandemic-related restrictions (i.e., lockdowns). Finally, grip strength was usually measured just before indoor air quality monitors were placed in participants’ homes. To assess whether this misalignment could have influenced our findings, we took advantage of having started outdoor air pollution measurements several weeks to months prior to grip strength assessment. We re-ran models using outdoor PM_2.5_ (1) averaged over all monitoring days preceding grip strength (mean = 22 days) and (2) averaged over 1, 7, or 30 day(s) prior to grip strength measurements (i.e., 1-day, 7-day, and 30-day lag) to address potential exposure misalignment.

All analyses were conducted using R version 4.3.1 [49].

## 3. Results

### 3.1. Participant Characteristics

The main analysis includes 2176 observations from 877 participants (648 in wave 1, 775 in wave 2, and 753 in wave 3). Participants’ socio-demographic characteristics were similar across waves (Appendix A). At baseline (i.e., wave 1), 61% of the study population was female, the mean age of which was 60.4 years with a standard deviation (SD) of 8.9. Most participants were married (86%) and had primary school education (73.9%), and over half (64.4%) worked in agriculture. Participants living in homes in the highest two quartiles of household PM_2.5_ had lower formal educational attainment and were more likely to work in agriculture or other manual labor (i.e., manufacturing, mining, or construction). They were also more likely to be a current smoker, consume alcohol daily, exercise less frequently, be exposed to higher levels of outdoor PM_2.5_, and have a higher maximum grip strength (Table 1).

### 3.2. Exposure to Household and Outdoor PM_2.5_

Wintertime (January 15 to March 15) average outdoor PM_2.5_ ranged from 8.3 to 100.1 μg/m^3^ across the study villages, with geometric mean concentrations of 43.8, 59.1, and 36.3 μg/m^3^ in waves 1, 2, and 4, respectively (Figure 1; Appendix A). Household PM_2.5_ was measured in a random sample of 227 households in wave 2 and 249 households in wave 4 across all 50 study villages. Seasonal averages ranged from 3 to 431 μg/m^3^ with geometric mean concentrations of 70.2 in wave 2 and 47 μg/m^3^ in wave 4 (Appendix A). The imputed household PM_2.5_ for all households had the same concentration range as measured household PM_2.5_, with geometric mean concentrations of 64, 67.7, and 52 μg/m^3^ in wave 1, 2, and 4, respectively.

### 3.3. Effect of Household and Outdoor PM_2.5_ Exposure on Maximum Grip Strength

Figure 2 illustrates how within- and between-individual changes in PM_2.5_ exposure relate to grip strength changes. We did not observe between-individual effects of either household or outdoor PM_2.5_ on grip strength in the multivariable models. For a 10 μg/m^3^ within-individual increase in household and outdoor PM_2.5_, the estimated changes in grip strength were 0.06 kg (95%CI: −0.01, 0.12 kg) and 1.51 kg (95%CI: 1.35, 1.68 kg), respectively, adjusted for covariates. After adjusting for outdoor PM_2.5_ in the household PM_2.5_ model, the change in grip strength following a 10 μg/m^3^ within-individual increase in household PM_2.5_ decreased to 0.01 kg (95%CI: −0.06, 0.08 kg), whereas the between-estimate remained unchanged.

We did not find evidence of effect measure modification by any variables evaluated for household PM_2.5_ on grip strength (Appendix A). For outdoor PM_2.5_, we only observed effect measure modification for between-individual effects, specifically, larger increases in grip strength for participants farming more frequently, potentially because this work tends to occur outdoors where participant exposure to outdoor PM_2.5_ is higher (Appendix A). We also found evidence of greater positive effects among older participants (i.e., ≤60 years versus >60 years, which is the common retirement age for adults in China) (Appendix A). However, these findings should be interpreted with caution given the wide confidence intervals for most interaction terms.

Complete case analyses with observed (non-imputed) data were consistent with results using the imputed data (Appendix A). Our results also did not change after including a random intercept for the participant’s village of residence, or additionally adjusting for co-morbid conditions, or indoor temperature measured during household visits, or clean energy implementation in participant’s living village (Appendix A). The results for household PM_2.5_ were similar after excluding the observations with the highest 3% and 5% of household PM_2.5_ observations, or excluding the observations measured less than 20% of the study period (Appendix A). Our results were also consistent in models limited to observed complete cases in waves 1 and 2 and in models that used the wintertime outdoor PM_2.5_ averaged to the period prior to grip strength measurement as the exposure (Appendix A).

## 4. Discussion

We used a longitudinal design and robust within-between effects approach to evaluate the effects of wintertime household (indoor) and outdoor PM_2.5_ on grip strength among middle-aged and older adults in rural Beijing. Overall, we did not find strong evidence of between-individual effects of either household or outdoor PM_2.5_ on grip strength. Contrary to expectations, we observed a 0.06 kg and 1.51 kg increase in maximum grip strength per 10 μg/m^3^ within-individual change in household and outdoor PM_2.5_ exposure, respectively. The household PM_2.5_ estimate attenuated to near zero after adjusting for outdoor PM_2.5_, but the inverse effects of outdoor PM_2.5_ on grip strength were robust to a large number of sensitivity analyses.

Our findings contrast with the very few previous studies of outdoor PM_2.5_ and grip strength. In France, an interquartile increase (3.9 μg/m^3^) in satellite-derived annual outdoor PM_2.5_ was associated with lower grip strength (−0.07 kg; 95%CI: −0.102, −0.05 kg) in urban adults [25]. Similarly, a multi-country study of six countries including China observed lower grip strength (−0.69 kg; 95%CI: −1.25, −0.12 kg) per 10 μg/m^3^ increase in satellite-derived annual PM_2.5_ in urban and peri-urban adults (mean age = 63 years) [24]. Several factors may explain the divergence between our findings and prior studies. First, both previous studies were cross-sectional, limiting their ability to establish temporality or rule out reverse causation. Second, the population-level estimates of exposure do not capture individual-level variation. In contrast, our within-individual estimates account for time-invariant confounders and reflect how changes in PM_2.5_ affect an individual’s grip strength over time. A population exposed to higher average air pollution may have lower average grip strength, but an individual experiencing short-term increases in PM_2.5_ may show transient, acute increases, which our study design is able to detect. Finally, the mean air pollution levels in our study (outdoor: 49 μg/m^3^; indoor: 80 μg/m^3^) are much higher than those reported in the previous two studies (6 and 32 μg/m^3^). It is possible that the PM_2.5_–grip strength relationship may differ at lower versus higher levels of exposure. However, we are unable to formally evaluate this in our study since less than 17% of participants had measured outdoor air pollution levels that fell within the ranges of previous studies.

Our study differs from several longitudinal studies in China among adults aged 45y+ that reported increased risk of sarcopenia, which is characterized by reduced physical performance and grip strength, associated with persistent solid fuel use for cooking and/or heating (hazard ratios: 1.26–1.56) [21,22,23]. However, these studies excluded over half of baseline participants due to missing data on fuel use, grip strength, or covariates, potentially introducing selection bias. As with studies using annual average outdoor PM_2.5_, household fuel type serves as a proxy for long-term household PM_2.5_ exposure, which may affect muscle function through mechanisms distinct from those of short-term exposure.

Our unexpected finding that short-term increases in PM_2.5_ increase grip strength within individuals most likely reflects transient physiological responses to air pollution. A possible mechanism is air pollution-induced endothelial dysfunction [50], which could alter vascular tone and lead to increased peripheral vascular resistance and elevated blood pressure [51]. Elevated blood pressure, previously associated with higher exposures to PM_2.5_ in rural China [52,53], can acutely support maximal force production and better grip strength, especially in older adults [54,55,56], where 10 mmHg increases in diastolic blood pressure were associated with 0.6 kg increases in grip strength for middle-aged adults (mean age = 63.2 years). However, these effects appear to be short-term, where a positive effect of blood pressure on grip strength shifted to an inverse relationship after several years of higher blood pressure [56]. Increased peripheral vascular resistance has greater potential to produce maximal grip strength, and the heightened sympathetic tone induced by air pollution may facilitate greater muscle activation [57]. PM_2.5_ may also stimulate the autonomic nervous system, which regulates muscle fibers and contraction [58,59,60], potentially improving short-term neuromuscular performance. Recent epidemiologic studies found that 2–4 month average exposure to outdoor PM_2.5_ decreased adiponectin levels [61,62,63], and reduced adiponectin levels have been paradoxically associated with better muscular fitness in older adults [64,65].

Still, any temporary and small improvements in grip strength do not diminish the well-documented health risks of PM_2.5_ and we do not interpret our study results as evidence that air pollution benefits overall physical function or slowing aging. Numerous epidemiologic and experimental studies link PM_2.5_ with systematic inflammation and the development of chronic disease, which may further result in muscle mass or strength loss [18,19,66] and decrease grip strength over the longer term [10]. Rather, our finding of a small increase in grip strength with higher seasonal outdoor PM_2.5_ highlights both the complexity of air pollution’s biological effects and the benefits of future mechanistic and experimental research that can differentiate between acute adaptive and long-term harmful pathways.

We found no evidence of between-individual effects, which may reflect the more limited variation in participant-level average (23 to 335 μg/m^3^) than within-individual changes (−222 to 374 μg/m^3^) in household PM_2.5_. However, for outdoor PM_2.5_, the variation in participant-level averages (22 to 92 μg/m^3^) is larger than the within-individual variation (−28 to 31 μg/m^3^). It could also indicate that our two-month PM_2.5_ measurement window is better suited for detecting short-term effects and that between-individual assessments require longer-term (e.g., annual) measurements.

We also observed stronger effects of outdoor PM_2.5_ compared with household PM_2.5_. This may be because of the differences in PM_2.5_ composition or source-specific toxicity. For example, sulphates that are more commonly attributed to outdoor sources such as traffic or factory emissions [67,68] are also associated with higher toxicity of airborne particles [69,70,71]. Studies in rural Beijing have shown that, although outdoor and indoor particles share sources, their physical and chemical properties can differ substantially [72,73]. However, even when particles originate from the same emissions source, the health impacts can differ by characteristics such as size, surface area, and chemical composition that are influenced by a range of factors including combustion efficiency and environmental conditions [74,75].

## 5. Strengths and Limitations

The strengths of our study compared with previous research on this topic include its longitudinal design, gold-standard measurement of grip strength, and the inclusion of a comprehensive set of individual-level covariates measured in each study wave such as exercise, socioeconomic status, and tobacco smoking. We applied within- and between-individual modeling approaches, which offer advantages over conventional regression adjustment by accounting for time-invariant unobserved confounders [40], and our two-month measurements of PM_2.5_ in homes and outdoors provide a more accurate representation of participants’ typical exposures in winter than the typical measurements of 24–72 h in settings of solid fuel use [76]. We directly measured wintertime household PM_2.5_, which provides a more accurate and temporally resolved assessment of indoor air quality than self-reported fuel use. Prior studies have primarily relied on self-reported household energy sources [24]. Further, fuel use is often strongly correlated with socioeconomic status [77], a key health determinant that was not adjusted for in previous studies [20,21], despite its known association with grip strength [78]. Additionally, our study design and use of multiple imputation allowed us to retain most of our sample and reduce any potential bias related to missing data on household exposures and covariates.

Our study also has several limitations to consider when interpreting our results and that could be considered in designing future studies. First, seasonal PM_2.5_ measurements began immediately after grip strength assessments due to study logistics. However, it is unlikely that participants’ indoor activities or stove use practices changed after grip strength measurement, and our sensitivity analyses using outdoor PM_2.5_ measured prior to grip strength assessments and considering different lag windows (e.g., 1-day, 7-day, or 30-day lag window) yielded similar results (Appendix A). Second, it is impossible to rule out residual confounding factors in any observational study, in our case particularly from time-varying unmeasured factors such as intensity (rather than frequency) of physical activity. However, the intensity of farming activities is unlikely to change over a short period of time (2 months). Third, excluding participants who were unable to complete grip strength measurements or participated in a single wave of data collection could introduce healthy participant bias. While there were small differences in social–demographic characteristics observed between these excluded participants and our sample (Appendix A), these exclusions are unlikely to have influenced outdoor PM_2.5_ and are therefore unlikely to bias our exposure–outcome estimates. Finally, our last wave of data collection coincided with the COVID-19 pandemic. Although no COVID-19 cases were reported in our study villages during the study period and all participants were subject to similar pandemic-related travel and distancing restrictions, it is possible that pandemic-related behavioral or environmental changes affected physical activity or air quality that are difficult to predict. However, restricting our analysis to the first two pre-pandemic waves did not change the between-individual effects of outdoor PM_2.5_ while the magnitude of the within-individual effect was slightly attenuated (Appendix A).

## 6. Public Health Implications and Future Research

Nearly half of the global population relies on solid fuels as their primary domestic energy source, particularly in low- and middle-income countries [1]. Our findings highlight the complexity of the impact of air pollution on physical functioning and muscle strength in settings where solid fuel use is widespread, providing novel insights for policymakers and public health practitioners. Future research may seek to further disentangle the acute and chronic effects of air pollution on functional decline, as well as the underlying mechanism. Longitudinal studies with repeated exposure and outcome measurements can help clarify potential temporal patterns and causal pathways of air pollution affecting physical functioning.

## 7. Conclusions

We did not observe between-individual effects of outdoor or household PM_2.5_ on grip strength among older adults in rural China, and the within-individual effects of household PM_2.5_ were attenuated after accounting for concurrent outdoor pollution levels. In contrast, unexpected positive effects of short-term within-individual increases in outdoor PM_2.5_ on grip strength were observed in the same participants and across many sensitivity analyses. While the mechanisms underlying these findings remain unclear, the results may reflect transient physiological responses to pollution and should not be interpreted as evidence of health benefits. These findings highlight the complexity of air pollution’s health impacts and the importance of longitudinal data for capturing time-sensitive effects. Further research could better understand these patterns and their implications for older adults in high-exposure environments.

## Figures and Tables

**Figure 1 ijerph-22-01283-f001:**
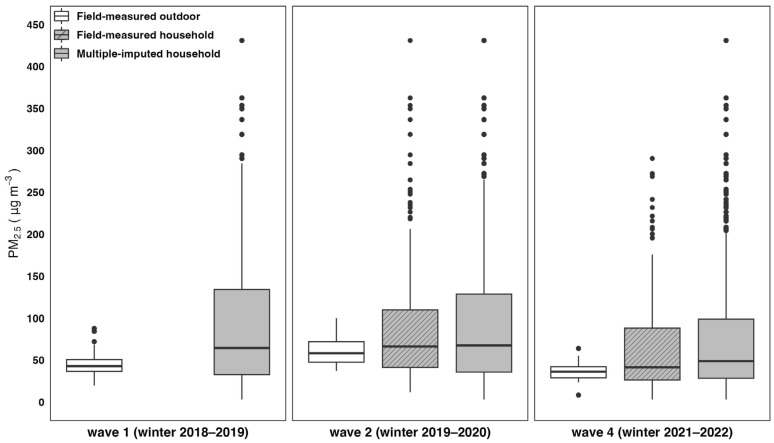
Wintertime average outdoor and household PM_2.5_ concentrations (μg/m^3^) from field measurements with multiple imputation.

**Figure 2 ijerph-22-01283-f002:**
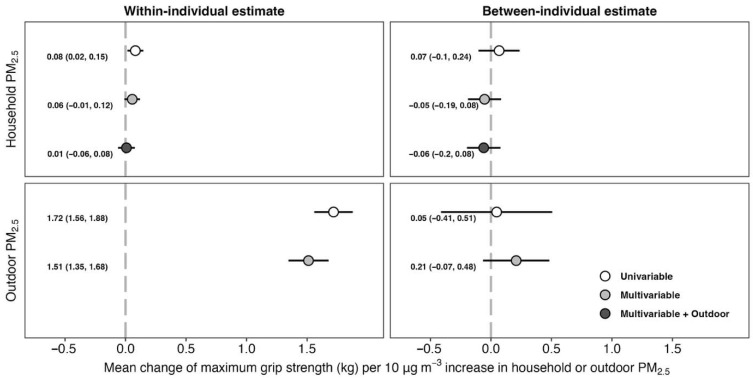
Change in maximum grip strength per 10 μg/m^3^ increase in wintertime household or outdoor PM_2.5_. ^a^ Multivariable models are adjusted for sex, age, marital status, highest education, current occupation, exposure to tobacco smoke, smoking intensity for current smokers, frequency of drinking, frequency of farming, exercise frequency, self-reported health status, waist circumference (with 2 degrees of freedom natural cubic spline), and asset-based wealth index quartile. ^b^ Multivariable + Outdoor models are adjusted for same variables as in Multivariable model plus wintertime average outdoor PM_2.5_ during same measurement period as household PM_2.5_.

**Table 1 ijerph-22-01283-t001:** Selected participant characteristics by quartile of wintertime average household PM_2.5_.

	Household (Indoor) PM_2.5_ Quartile (μg/m^3^)				
	Overall	≤25th %	25th to 50th %	50th to 75th %	>75th %
		3 < Household ≤ 32	32 < Household ≤ 61	61 < Household ≤ 120	120 < Household ≤ 431
Age, years	61.8 (9.0)	61.8 (9)	62.1 (8.9)	61.8 (8.9)	61.6 (9.1)
Sex, % female	60.6	61.5	61.7	62.6	56.6
Marital status					
Married	87.1	88.4	88.1	87.0	85.3
Divorced or separated	1.9	0.8	1.1	1.8	3.7
Widowed	10.0	10.0	10.1	10.4	9.4
Never married	1.0	0.8	0.8	0.8	1.6
Highest education					
No school	11.8	12.6	12.3	11.2	10.9
Primary school	75.7	73.3	73.7	76.8	78.8
Secondary or high school	11.2	11.6	12.6	11.0	9.7
Higher education	1.3	2.5	1.3	0.9	0.6
Current occupation					
Agriculture	58.8	54.7	56.3	59.8	64.5
Other manual labor ^a^	1.8	1.0	1.9	2.0	2.4
Non-manual labor ^b^	7.5	9.9	8.7	7.1	4.2
Unemployed	26.0	27.9	26.2	25.2	24.7
Others	5.9	6.5	7.0	5.8	4.3
Tobacco smoke					
Current smoker	25.0	14.0	17.4	25.1	43.2
Former smoker	14.3	18.3	17.1	13.2	8.6
Never smoker lived with smoker	41.8	40.1	41.3	45.5	40.2
No exposure to tobacco smoke	19.0	27.6	24.2	16.2	8.0
Frequency of drinking					
Never	50.6	49.5	51.5	52.9	48.5
Occasional (≤3 times a month)	21.7	22.9	20.5	20.5	22.7
Regular (≤5 times a week)	7.9	9.6	9.4	7.5	5.3
Everyday	19.8	18.1	18.6	19.0	23.5
Frequency of farming					
Never	40.5	40.4	40.4	40.7	40.6
Occasional (≤2 days a week)	30.1	30.6	29.1	31.2	29.6
Regular (≤5 days a week)	18.9	20.5	20.5	18.3	16.3
Everyday	10.5	8.6	10.0	9.8	13.6
Frequency of exercising					
Never	21.9	18.2	21.3	21.6	26.3
Occasional (≤2 days a week)	14.1	16.2	14.8	13.3	12.0
Regular (≤5 days a week)	10.8	11.3	11.0	11.1	9.9
Everyday	53.2	54.3	52.8	54.0	51.8
Self-reported health					
Excellent	3.4	3.4	2.8	3.3	4.2
Good	43.0	46.9	43.4	41.8	39.9
Fair	17.2	15.3	16.8	18.4	18.4
Poor	36.4	34.4	37.0	36.5	37.6
Asset-based wealth index ^c^					
Bottom quartile (poorest)	24.5	25.1	24.6	24.1	24.4
2nd quartile	24.7	24.7	24.5	25.3	24.2
3rd quartile	25.7	25.0	25.0	25.3	27.3
Top quartile (wealthiest)	25.1	25.2	25.8	25.2	24.1
BMI, kg/m^2^	26.0 (3.6)	26 (3.5)	26 (3.6)	26 (3.8)	25.9 (3.6)
Waist circumference, cm	88.5 (10.1)	88.6 (10.0)	88.5 (10.3)	88.3 (10.5)	88.6 (9.8)
Outdoor PM_2.5_, μg/m^3^	45.6 (1.4)	43.3 (1.4)	45.7 (1.4)	46.8 (1.5)	46.8 (1.4)
Maximum grip strength, kg	31.1 (10.3)	30.9 (10.5)	30.8 (10.5)	30.6 (9.9)	31.9 (10.3)

Note: Percentages are given for categorical variables, mean (standard deviation) are given for continuous variables. Geometric mean (geometric standard deviation) are given for outdoor PM_2.5_ considering its skewed distribution. ^a^ Other manual labor includes manufacturing, mining, and construction workers. ^b^ Non-manual labor includes government, technical, and professional service workers. ^c^ Created based on ownership of household assets using principal component analysis. Details can be found in Li et al. [33].

## Data Availability

The datasets used and/or analyzed in this study are available from the corresponding author on reasonable request. The codes used for this study are available in the repository: https://osf.io/hbtjs/?view_only=1eedb15e28bf4868b5cc708f8fffd104 (accessed on 1 August 2025).

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
