# Peer review of "Effects of Outdoor and Household Air Pollution on Hand Grip Strength in a Longitudinal Study of Rural Beijing Adults"

_ijerph, 2025, doi:10.3390/ijerph22081283_

Round 1
Reviewer 1 Report
Comments and Suggestions for Authors
This manuscript presents an important and well-executed analysis examining the association between seasonal PM2.5 exposure and grip strength among older adults in rural China, using repeated measurements and mixed-effects modeling. The topic is timely and of public health relevance, particularly in the context of aging populations and environmental health risks. The use of within- and between-individual modeling is a notable strength, as is the effort to incorporate both indoor and outdoor pollution data.
However, several aspects of the study design, statistical modeling, and interpretation could be strengthened to improve the clarity, causal inference, and alignment with the study aims. Detailed comments are provided below:
Introduction:
While the review of past literature is thorough, a clearer summary of mechanisms linking PM2.5 to muscle function (e.g., inflammation, oxidative stress) could help reinforce the rationale.
Methods & Results:
1. Participants’ grip strength was measured first, and then the indoor and outdoor PM2.5 monitors were installed for 2–3 months. This creates a temporal misalignment: the measured exposure may not reflect the pollution levels that influenced the participant's physical function at the time of grip strength testing.
2. Add basic diagnostic plots (Q–Q plots, residual vs. fitted, leverage) or reference them in supplemental material to confirm model assumptions hold.
3. The use of a 22-day average lag for outdoor PM2.5 is a reasonable sensitivity check given the timing mismatch, but the rationale appears to be logistical rather than biologically driven. The authors could consider discussing the limitations of this fixed lag period and explore whether different lag windows (e.g., 7-day, 30-day) or distributed lag models might provide additional insights into short-term vs. delayed effects
4. Test whether random slopes for PM2.5 improve model fit (i.e., individuals vary in how PM2.5 affects them). Perform a likelihood ratio test to compare model with and without random slopes.
5. Highlight significant interactions more clearly and consider plotting marginal effects (e.g., PM2.5 effect by farming frequency or age group).
6. Examine change in grip strength over time as a continuous outcome.
Discussion:
The discussion would benefit from a more balanced and critical interpretation of the findings. This includes better acknowledging the discrepancy between study aims and key results, the implications of measuring exposure after the outcome, and a more detailed exploration of potential explanations for the unexpected within-individual positive association. Expanding the limitations, situating findings within a broader literature base, and offering clearer future research directions would enhance the paper’s contribution
Reviewer 2 Report
Comments and Suggestions for Authors
this is an interesting study within a wider longitubdinal dataset looking at a marker of aging
Overall this is a well written article and I have no concerns regarding the language or grammar.
My main comments is around the pollution data - as grip strength is a marker of aging - it would be important to have background information on exposures as well as the acute ones included in the study. Do you have the information regarding exppsures prior to the study starting? or a surrogate marker for this?
Author Response
Please see the attachment.
The responses to Reviewer 2 are under the section Reviewer 2.

Reviewer 3 Report
Comments and Suggestions for Authors
Reviewer Comment
Manuscript Title: Effects of Outdoor and Household Air Pollution on Hand Grip Strength in a Longitudinal Study of Rural Beijing Adults
- General Comments
This manuscript presents a well-executed longitudinal study on the associations between indoor and outdoor PM2.5 exposures and hand grip strength among middle-aged and older adults in rural Beijing. The study fills a notable gap in environmental health literature by focusing on a high-exposure, aging population, and by using repeated measures and direct PM2.5 monitoring.
The use of within-between subject decomposition models, multiple imputations for missing exposure data, and robust sensitivity analyses add to the methodological strength of the work. The unexpected positive association between outdoor PM2.5 and grip strength is an interesting finding that merits careful interpretation.
To strengthen the manuscript, I suggest clarifying issues related to the sampling design, clinical relevance of findings, and the interpretation of unexpected results. Below are specific, actionable suggestions.
- Specific Comments
2.1 Abstract
- The abstract demonstrates clarity and detail. Nevertheless, the unanticipated positive correlation between outdoor PM2.5 and grip strength could be misconstrued as advantageous. It is recommended to highlight the necessity for careful interpretation in the concluding remarks.
- In line 44-46, the statement “these findings should not be interpreted as evidence that air pollution benefits aging,.....”. This sentence is appropriate and necessary, but should consider rephrasing to underscore the need for further investigation without suggesting that the readers might misinterpret.
2.2 Introduction
- Flow of Citations: While citations are present, ensuring they are consistently placed at the end of sentences or clauses where the information is derived would enhance readability. For example, “[6]” appear mid-sentence in line 61, which can be slightly disruptive.
- Specificity on "Early Indicators": The statement "identifying early indicators of functional decline that may be influenced by environmental exposures like air pollution” in lines 59-60, could be slightly expanded to briefly explain why early indicators are crucial in this context (e.g., for timely interventions or prevention).
2.3 Methods – Study Design
- In lines 115-128, the shift from random sampling to participant referrals could introduce selection bias. Please clarify whether baseline characteristics differed between early and later recruits.
- In lines 112 and 131, “Supplymental Material “should be corrected to “Supplementary Material”
- In line 129, the phrase “participants that had” should be revised to “participants who had”
- In line 134-137, the statement noted that sub-sample of 300 households were randomly selected for indoor PM₂.₅ monitoring, but the rationale and method of this subsampling strategy is not clearly explained. Please specify how the random selection was conducted (e.g., stratified by village or participant characteristics), and clarify whether the subsample was intended to be representative of the full study population. This information is important to assess the potential bias and generalizability of indoor exposure estimates.
- In Section 2.3.2, Measurement of outdoor (community) PM₂.₅: Explicitly state the duration of continuous measurement for outdoor PM2.5 monitoring during each wave to enhance clarity and reproducibility.
2.4 Statistical Analysis
- To enhance the understanding of a wider audience, including policymakers and health practitioners, we recommend the authors to consider adding a more accessible explanation of the 'within-individual' and 'between-individual' effects in Section 2.7, 'Statistical analysis.'
2.5 Results
- Detailed Description of Figure 2: Although Figure 2 (the forest plot) is referenced, a more comprehensive textual elucidation of its visual implications, exceeding the mere presentation of coefficients and confidence intervals, would be advantageous. For instance, explicitly articulating how the plot delineates the attenuation of household PM2.5 effects after adjusting for outdoor PM2.5 would serve to reinforce the visual interpretation.
2.6 Discussion
Overall, the discussion section is a strong component of the paper, particularly in its rigorous interpretation of the unexpected positive effects.
- Limitations Section Structure: The integration of limitations throughout the text may hinder quick identification of study constraints, suggesting the need for a dedicated subsection. Clear signposting of significant limitations is crucial if they remain integrated.
- Future Research Directions: A structured section explicitly outlining future research directions could enhance the guidance for subsequent studies and highlight research gaps.
- Broader Public Health Implications: The discussion should elaborate on the broader public health implications of findings, particularly regarding the relationship between acute effects and chronic air pollution exposure, as well as their impact on public health strategies.
The manuscript is clearly structured and written. I suggest minor edits to improve flow and simplify complex methodological language.
Author Response
Please see the attachment.
The responses to Reviewer 3 are under the section Reviewer 3.

Round 2
Reviewer 1 Report
Comments and Suggestions for Authors
Thank you for addressing all previous comments. The revisions have improved the manuscript, and I have no further suggestions.